# Musculoskeletal misdiagnoses in children with brain tumors: A nationwide, multicenter case-control study

**Laura Hallundbæk**[1], **Søren Hagstrøm**[1], **Rene Mathiasen**[2], **Troels Herlin**[3], **Henrik Hasle**[3], **Kathrine Synne Weile**[3], **Jesper Amstrup**[1], **Ninna Brix**[1]*

**1** Department of Pediatric and Adolescent Medicine, Aalborg University Hospital, Aalborg, Denmark,
**2** Department of Pediatric and Adolescent Medicine, Rigshospitalet, Copenhagen, Denmark, **3** Department of Pediatric and Adolescent Medicine, Aarhus University Hospital, Aarhus, Denmark

* ninna.brix@rn.dk

**Data Availability Statement:** All relevant data are within the paper and its Supporting Information files.

## Abstract

### Objective

Childhood brain tumors belong to the cancer type with the longest diagnostic delay, the highest health care utilization prior to diagnosis, and the highest burden of long-term sequelae. We aimed to clarify whether prior musculoskeletal diagnoses in childhood brain cancer were misdiagnoses and whether it affected the diagnostic delay.

### Study design

In this retrospective, chart-reviewed case-control study we compared 28 children with brain tumors and a prior musculoskeletal diagnosis to a sex and age-matched control group of 56 children with brain tumors and no prior musculoskeletal diagnosis. Using the Danish registries, the cases were identified from consecutive cases of childhood brain cancers in Denmark over 23 years (1996–2018).

### Results

Of 931 children with brain tumors, 3% (28/931) had a prior musculoskeletal diagnosis, of which 39% (11/28) were misdiagnoses. The misdiagnoses primarily included torticollis-related diagnoses which tended to a longer time interval from first hospital contact until a specialist was involved: 35 days (IQR 6–166 days) compared to 3 days (IQR 1–48 days), p = 0.07. When comparing the 28 children with a prior musculoskeletal diagnosis with a matched control group without a prior musculoskeletal diagnosis, we found no difference in the non-musculoskeletal clinical presentation, the diagnostic time interval, or survival. Infratentorial tumor location was associated with a seven-fold risk of musculoskeletal misdiagnosis compared to supratentorial tumor location.

**Funding:** The study was funded by the Arvid Nilsson's foundation(1780631), Børnecancerfonden (2017-1945 and 2020-6653), Ølufgard Memorial Fund (25734), and Aarhus University. No sponsors or funders (other than the named authors) played any role in study design, data collection and analy-sis, decision to publish or preparation of the manuscript.

**Competing interests:** The authors have declared that no competing interests exist.

**Abbreviations:** CNS, Central nervous system; DCCR, Danish Childhood Cancer Registry; DNPR, The Danish National Patient Registry; IQR, interquartile range.

## Conclusion

Musculoskeletal misdiagnoses were rare in children with brain tumors and had no significant association to the diagnostic time interval or survival. The misdiagnoses consisted primarily of torticollis- or otherwise neck-related diagnoses.

## Introduction

Among childhood cancers, brain tumors are the most common type of solid tumors and the leading cause of death [1,2]. The overall survival rate has shown improvements over the last decade, now approaching a five-year survival of 83% in Denmark [3]. However, it is still below the average of 86% for other childhood cancers [4]. Pediatric brain tumors are considered among the cancer types with the longest diagnostic delay and the highest healthcare utilization prior to diagnosis [5–9]. Further, survivors of brain tumors face the highest disease burden of long-term effects with substantial morbidity, including increased risk of subsequent tumors and neurocognitive deficits, markedly reducing their quality of life [10–14].

Brain tumors have rarely been described with musculoskeletal symptoms other than gait disturbances or neck pain [15–18]. Though, considering the whole spectrum of childhood cancers, musculoskeletal symptoms at debut have been found in up to 25% and musculoskeletal diagnoses in 7–12% [19–21]. Musculoskeletal misdiagnoses have primarily been described in children with leukemia and lymphomas [19,22], and only a few cases of musculoskeletal misdiagnoses in children with brain tumors have been described [21,23,24]. A recently published nationwide registry-based cohort study by our group, including all children with cancer in Denmark over 23 years, identified a musculoskeletal diagnosis prior to the diagnosis of cancer in 4% (33/931) with brain tumors, [20].

In the present study, the children registered with prior musculoskeletal diagnoses (cases) were compared to age, sex, and cancer-matched controls as we aimed to evaluate if they differed regarding the clinical presentation, diagnostic delay, and survival. Secondarily, we aimed to clarify whether the prior musculoskeletal diagnoses in children with brain tumors were misdiagnoses and aimed to identify any patterns or red flags.

## Materials and methods

### Study design

We performed a nationwide, multicenter case-control study collecting data by chart review. The cohort was identified from a nationwide (population of 5.8 million) registry-based cohort study, previously described in detail [20]. All children 0–14 years of age diagnosed with brain tumors in Denmark from January 1, 1996, to December 31, 2018, were identified using The Danish Childhood Cancer Registry (DCCR). The Danish National Patient Registry (DNPR) was used to identify musculoskeletal diagnoses and associated dates recorded within six months preceding the diagnosis of the brain tumor. The children with a prior musculoskeletal diagnosis (cases) were each compared to two sex- and age-matched (+/- 1 year) children with brain cancer but without registered musculoskeletal diagnoses (controls). The musculoskeletal diagnoses were defined as diagnoses from ICD-10 Chapter XIII (diseases of the musculoskeletal system and connective tissue), M00-M99, and the R-diagnosis "Pain, not elsewhere classified" from chapter XVIII (Symptoms, signs, and abnormal clinical and laboratory findings,

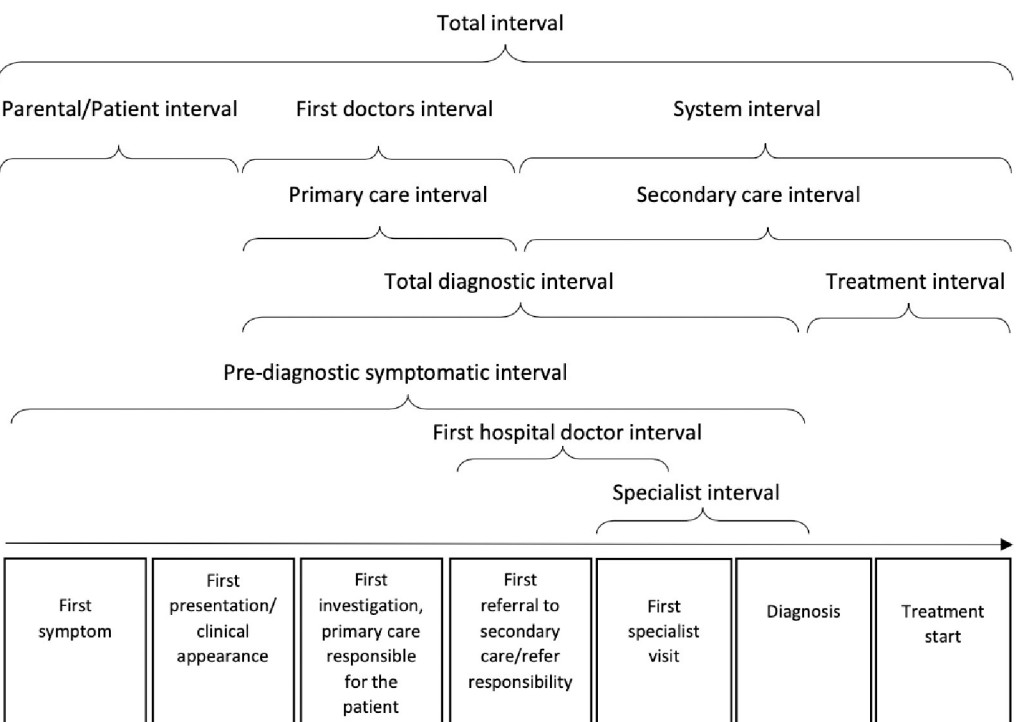

**Fig 1. Definition of time intervals.** An illustration of the overall milestones and time intervals in the route from the first symptom until the start of treatment.

not elsewhere classified). The diagnoses included were from both hospitalizations, emergency room, and outpatient visits but not from general practitioners [20].

We used a standardized definition of time intervals, including the route from the first symptom until the start of treatment [25]. In addition, three additional time intervals were added: 1) a pre-diagnostic symptomatic interval, i.e., the time from the first symptom until diagnosis. 2) a first hospital doctor interval, i.e., the time from the first hospital contact until a specialist was involved; and 3) a specialist interval, i.e., the time from the specialist was involved until the final cancer diagnosis was made (the specialist was defined as either a pediatric oncologist or neurosurgeon). The time intervals are illustrated in Fig 1.

## Data collection

Data were collected by reviewing medical charts from all pediatric departments in Denmark. The following data were collected: age, sex, cancer diagnosis, date of diagnosis, musculoskeletal diagnosis, tumor location and type, metastases, cancer treatment, presence of a musculoskeletal misdiagnosis, medical mistreatment, comorbidities, cause of death, and date of death, last day of follow-up and organ specified sequelae related to cancer and treatment. Further, we collected information on symptoms, objective signs, laboratory values, imaging, and time intervals (including the predefined time intervals). Musculoskeletal misdiagnosis was defined as a musculoskeletal diagnosis causing symptoms later explained by the cancer diagnosis. The date of diagnosis was defined as the day a brain tumor was identified on imaging. If the children with a musculoskeletal diagnosis no longer had musculoskeletal symptoms present at the time of brain cancer diagnosis, the musculoskeletal symptoms were not included in the data collection and the musculoskeletal diagnoses were defined as non-misdiagnoses.

## Cases and controls

From January 1996 to December 2018, 931 children aged 0–14 years were diagnosed with brain tumors in Denmark, identified using the Danish Childhood Cancer Registry. Hereof, 33/931 (4%) were registered with a musculoskeletal diagnosis within six months before the diagnosis of the brain tumor, according to data from DNPR. Four patients were excluded during the evaluation of medical charts due to DNPR misclassification and one due to missing records. The positive predictive value of a prior musculoskeletal diagnosis in childhood brain tumors identified from musculoskeletal diagnosis in DNPR was 85% (28/33). The 28 cases with brain tumors and a prior musculoskeletal diagnosis were randomly matched with two controls with brain tumors without musculoskeletal ICD diagnosis having the same sex and age (+/- 1 year) at the time of the brain tumor diagnosis Fig 2.

## Statistical analysis

All continuous data were non-normally distributed (evaluated by histograms and QQ-plots), and comparisons were made by Mann Whitney U-test and tabulated with median and inter-quartile range (IQR). Categorical data were tabulated by prevalence, and Fisher's exact test was used for comparisons. For mortality analysis, we followed patients from the date of cancer diagnosis until death, emigration, or the end of follow-up (September 1, 2021), whichever came first. The Kaplan-Meier method was used to compare the one-and five-year overall survival for the children with a prior musculoskeletal diagnosis versus controls. Logistic regression was used to determine factors associated with a musculoskeletal misdiagnosis. All statistical tests were performed under a two-sided significance level of 0.05. We used STATA V. 17.0 (Stata MP) for the statistical analysis.

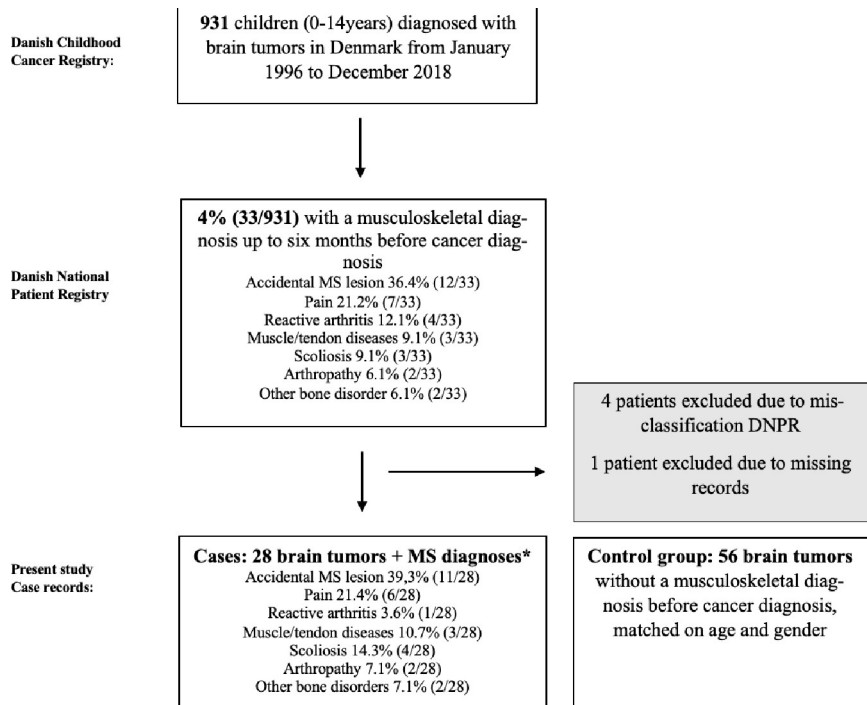

**Fig 2. Flow chart of the study population.** Flow chart illustration of the total cohort, and the final case group of children with brain tumors and a prior musculoskeletal diagnosis.

## Ethics

The Danish Data Protection Agency approved the study on August 31, 2021 (1-45-70-90-21). The reporting of the study was performed according to STROBE, *Strengthening the Reporting of Observational Studies in Epidemiology*, recommendations for case-control studies. For this type of study formal consent is not required.

## Results

We compared the 28 children with brain tumors and a prior musculoskeletal diagnosis (cases) to the 56 age- and gender-matched children with brain tumors without a prior musculoskeletal diagnosis (controls), first regarding the tumor characteristics, treatment, sequelae, and death, Table 1. Median age and gender did not differ for the cases versus controls. For the total cohort, the median age was seven years (IQR 4–11) and 46% (39/84) were girls. The prevalence of the cases and controls did not differ when dividing into the two periods, as 64% of the cases and 61% of the controls were diagnosed in the period 1996–2006 (p = 0.75) and 39% cases and 36% controls were diagnosed from 2007–2018 (p = 0.75). The predominant type of brain tumors was astrocytoma (24%) and medulloblastomas (23%), and 56% of the tumors were malignant, Table 1. The cases and controls did not differ regarding tumor location, type, or grade of tumor, and the treatment modalities were evenly distributed. Also, metastases and comorbidity were equally frequent, Table 1. Though, sequelae were more frequent among the controls 78% (42/54) than in the cases 50% (13/26), p = 0.01, Table 1. Sequelae included both sequelae due to treatment and the tumor. The organ-specific location of the sequelae did not differ between the two groups, Table 1. The median follow-up time did not vary for the cases: 13 years (IQR 8–18) and controls: 11 years (IQR 5–17), p = 0.51. We found no difference in overall survival comparing the cases and the controls as illustrated in the Kaplan-Meier plot Fig 3.

The cases and controls did not differ regarding first presenting symptoms, neurological symptoms, neurological findings, general symptoms, or the number of affected symptoms Table 2. The most frequent neurological findings were impaired motor or sensory function (17%) and ataxia (12%), Table 3. At the time of the diagnosis of the brain tumor, 14% (8/56) of the controls had musculoskeletal symptoms (though not registered in DNPR) compared to 48% (13/28) of the cases, p = 0.002, Table 2. Thereby, half of the cases did no longer have musculoskeletal symptoms at the time of cancer diagnosis and their previous musculoskeletal diagnoses were presumably non-related to the cancer as it either disappeared spontaneously or responded to non-cancer treatment. Torticollis-related symptoms as the first presenting symptom occurred in 14% (4/28) of the cases compared to 2% (1/56) of the controls, p = 0.02. The musculoskeletal symptom mainly included neck pain and gait abnormalities. Arthralgia, arthritis, nocturnal pain, or morning stiffness did not occur in any of the groups.

For the analysis of time intervals, six children with a prior diagnosis of neurofibromatosis and one child with tuberous sclerosis prior to the brain tumor diagnosis were excluded, as their comorbidity resulted in routine surveillance imaging and thereby short total time intervals with a median of 0 days (IQR 0–8). The total diagnostic interval did not differ for the cases versus the controls, as illustrated in Fig 4. A total time interval of more than six months occurred in 33% (8/24) of the cases and 42% (22/52) of the controls, p = 0.61. The parental and primary care interval accounted for the most significant proportion of the interval being 38 days (IQR 14–174) for the cases and 77 days (IQR 17–273) for the controls, p = 0.65. The first hospital doctor interval had wide ranges but a median of just six days (IQR 1–54) for the cases and four days (IQR 1–68) for the controls.

**Table 1. The prevalence of the clinical characteristics in the total cohort and the children with brain tumors and a prior musculoskeletal diagnosis (cases) versus age- and gender-matched children with brain tumors without a prior musculoskeletal diagnosis (controls).**

| | Total cohort N = 84 | Cases N = 28 | Controls N = 56 | P |
|---|---|---|---|---|
| **Tumor location** | | | | |
| Supratentorial tumor | 55% (46/84) | 50% (14/28) | 58% (32/55) | 0.64 |
| Infratentorial tumor | 45% (37/84) | 50% (14/28) | 42% (23/55) | 0.48 |
| **Tumor grade** | | | | |
| Malignant | 56% (36/64) | 64% (14/22) | 52% (22/42) | 0.45 |
| Benign | 44% (28/64) | 36% (8/22) | 48% (20/42) | 0.63 |
| **Tumor type** | | | | |
| Astrocytoma grade 1–2 | 24% (20/84) | 21% (6/28) | 25% (14/56) | 0.72 |
| Astrocytoma grade 3–4 | 6% (5/84) | 7% (2/28) | 5% (3/56) | 0.74 |
| Other gliomas | 8% (7/84) | 11% (3/28) | 7% (4/56) | 0.58 |
| Medulloblastoma | 23% (19/84) | 29% (8/28) | 20% (11/56) | 0.26 |
| Glioblastoma | 1% (1/84) | 0% (0/28) | 2% (1/56) | 0.48 |
| Meningioma | 1% (1/84) | 4% (1/28) | 0% (0/56) | 0.15 |
| Ependymoma | 2% (2/84) | 0% (0/28) | 4% (2/56) | 0.31 |
| Germinal cell tumor | 2% (2/84) | 0% (0/28) | 4% (2/56) | 0.31 |
| Craniopharyngioma | 7% (4/84) | 0% (0/28) | 5% (4/56) | 0.15 |
| Other tumors | 12% (10/84) | 14% (4/28) | 11% (6/56) | 0.63 |
| Unspecified tumor | 17% (14/84) | 14% (4/28) | 18% (10/56) | 0.68 |
| **Metastases** | **5% (4/83)** | **4% (1/28)** | **6% (3/55)** | **0.70** |
| **Comorbidity** | **29% (24/84)** | **39% (11/28)** | **23% (13/56)** | **0.12** |
| **Treatment** | | | | |
| Surgery | 75% (63/84) | 79% (22/28) | 73% (41/56) | 0.59 |
| Radical surgery | 47% (29/62) | 59% (13/22) | 40% (16/40) | 0.15 |
| Radiotherapy | 43% (36/84) | 39% (11/28) | 45% (25/56) | 0.64 |
| Chemotherapy | 46% (39/84) | 43% (12/28) | 48% (27/56) | 0.64 |
| Other primary treatment | 27% (22/83) | 29% (8/28) | 26% (14/55) | 0.76 |
| Watchful waiting | 17% (14/84) | 21% (6/28) | 14% (8/56) | 0.29 |
| **Death related to cancer/side-effects** | **21% (18/84)** | **18% (5/28)** | **23% (13/56)** | **0.57** |
| **Sequelae related to treatment or cancer** | **70% (55/80)** | **50% (13/26)** | **78% (42/54)** | **0.01** |
| Central nervous system sequelae | 48% (40/84) | 36% (10/28) | 54% (30/56) | 0.12 |
| Cardiopulmonary sequelae | 0% (0/84) | 0% (0/28) | 0% (0/56) | - |
| Gastrointestinal sequelae | 1% (1/84) | 0% (0/28) | 2% (1/56) | 0.48 |
| Musculoskeletal sequelae | 25% (21/84) | 14% (4/28) | 30% (17/56) | 0.11 |
| Other sequelae | 1% (1/84) | 0% (0/28) | 2% (1/56) | 0.48 |

Other primary treatments included steroids and pressure-relieving operations. Comorbidity was defined as any chronic illness described in the medical charts prior to diagnosis and included illnesses such as asthma and neurofibromatosis. Unspecified tumors included tumors without biopsies to determine the tumor type.

Misdiagnosis occurred in 39% (11/28) of the cases, corresponding to an estimated prevalence of musculoskeletal misdiagnoses in childhood brain tumors of 1% (11/931). Medical mistreatment did not occur in any of the cases. Torticollis-related diagnoses constituted 64% (7/11) and 36% (4/11) were pain-related. Five children were diagnosed (specifically) with torticollis at the age of eight months, one, two, three, and four years, respectively. The remaining two were misdiagnosed with *sequelae after superficial lesion* and *muscle strain* of the neck at the age of four and seven years. The pain-related misdiagnoses included *musculoskeletal pain*, *leg pain*, *back pain*, and *unspecified pain*.

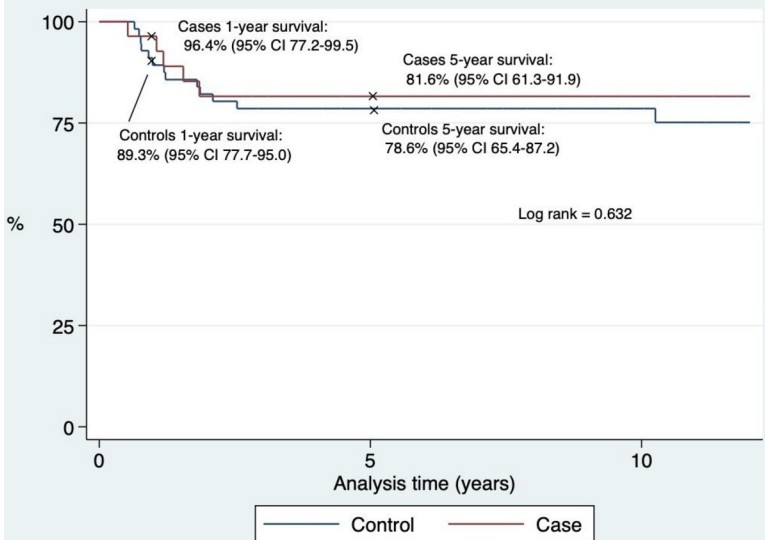

**Fig 3. Survival rates.** Kaplan-Meier curve of the survival rates of the children with a brain tumor, and a prior musculoskeletal diagnosis (case) versus those with a brain tumor without a prior musculoskeletal diagnosis (control).

The 11 children with a musculoskeletal misdiagnosis were compared to the children without any musculoskeletal misdiagnosis (n = 73), including the 17 children with a prior musculoskeletal diagnosis unrelated to the brain tumors, Table 4. The diagnostic interval did not differ for the two groups. The children with a musculoskeletal misdiagnosis had a total interval of 169 days (IQR 57–582) compared to 86 days (IQR 32–397) for the children without a musculoskeletal misdiagnosis, p = 0.38. The interval of the first hospital doctor tended to be longer for children with a musculoskeletal misdiagnosis (35 days (IQR 6–166)) compared to 3 days (IQR 1–48) for the group without a musculoskeletal misdiagnosis, p = 0.07, Table 4. A total interval above six months occurred in 46% (5/11) of the children with a musculoskeletal misdiagnosis and 39% (25/65) without a musculoskeletal misdiagnosis, p = 0.45.

Finally, we controlled for factors associated with a musculoskeletal misdiagnosis and found a seven-fold increased risk of musculoskeletal misdiagnosis in the case of an infratentorial tumor location compared to a supratentorial tumor location, OR 7.1 (95% CI 1.4–35.2). There were no significant associations between musculoskeletal misdiagnosis and tumor type, time interval, neurological symptoms, age, sex, or comorbidity. Though, reservations must be made for sub-analyses carried out on the small numbers.

## Discussion

Of 931 children with brain tumors, 3% had a musculoskeletal diagnosis in a period of up to six months prior to the diagnosis of the brain tumor. Hereof 39% (11/28) were misdiagnoses, two-thirds as torticollis-related diagnoses and one-third as pain-related diagnoses. This subgroup tended to have a longer interval from the first hospital contact until a specialist was involved.

In several studies, head tilt has been revealed as a symptom of brain tumors, with a prevalence of between 5–11% [15,16,26–30]. A smaller study, including 50 children with fossa posterior tumors, revealed that torticollis-related symptoms often were misinterpreted [31]. Further, few studies have found an increased risk of diagnostic delay in case of torticollis or head tilt as presenting symptoms [28,32]. In children above one year of age, torticollis should

**Table 2. The prevalence of symptoms, and abnormal laboratory values in the total cohort, the children with brain tumors and a prior musculoskeletal diagnosis (cases) versus the children with brain tumors without a prior musculoskeletal diagnosis (controls).**

| | Total N = 84 | Cases N = 28 | Controls N = 56 | **P** |
|---|---|---|---|---|
| **First presenting symptom** | | | | |
| Seizure | 10% (8/84) | 11% (3/28) | 9% (5/56) | 0.79 |
| Neck pain | 2% (2/84) | 4% (1/28) | 2% (1/56) | 0.61 |
| Vomiting | 7% (6/84) | 7% (2/28) | 7% (4/56) | 1.00 |
| Headache | 27% (23/84) | 25% (7/28) | 29% (16/56) | 0.73 |
| Torticollis-related symptoms | 6% (5/84) | 14% (4/28) | 2% (1/56) | 0.02 |
| Visual difficulties | 10% (8/84) | 11% (3/84) | 9% (5/56) | 0.79 |
| Impaired sensory or motor function | 10% (8/84) | 4% (1/28) | 13% (7/56) | 0.19 |
| Loss of skills | 4% (3/84) | 7% (2/28) | 2% (1/56) | 0.21 |
| Growth deficiency | 6% (5/84) | 0% (0/28) | 9% (5/56) | 0.10 |
| No symptoms | 7% (6/84) | 11% (3/28) | 5% (3/56) | 0.37 |
| Other symptoms | 12% (10/84) | 7% (2/28) | 14% (8/56) | 0.34 |
| **Musculoskeletal symptoms** | **26% (21/84)** | **48% (13/28)** | **14 (8/56)** | **0.002** |
| Leg pain | 1% (1/84) | 4% (1/28) | 0% (0/56) | 0.15 |
| Gait abnormalities | 10% (8/84) | 14% (4/28) | 7% (4/56) | 0.29 |
| Back pain | 2% (2/84) | 7% (2/28) | 0% (0/56) | 0.04 |
| Neck pain | 8% (7/84) | 21% (6/28) | 2% (1/56) | 0.005 |
| Arthralgia | 0% (0/84) | 0% (0/28) | 0% (0/56) | - |
| Arthritis | 0% (0/84) | 0% (0/28) | 0% (0/56) | - |
| Nighttime pain | 0% (0/84) | 0% (0/28) | 0% (0/56) | - |
| Morning stiffness | 0% (0/84) | 0% (0/28) | 0% (0/56) | - |
| Unspecified musculoskeletal symptoms | 10% (8/84) | 18% (5/28) | 5% (3/56) | 0.07 |
| **Neurological symptoms** | **83% (70/84)** | **82% (23/28)** | **84% (47/56)** | **0.84** |
| Headache | 49% (41/84) | 46% (13/28) | 50% (28/56) | 0.76 |
| Dizziness | 16% (13/84) | 21% (6/28) | 13% (7/56) | 0.29 |
| Vomiting | 44% (37/84) | 39% (11/28) | 46% (26/56) | 0.53 |
| Seizure | 12% (10/84) | 11% (3/28) | 13% (7/56) | 0.81 |
| Paresis | 11% (9/84) | 7% (2/28) | 13 (7/56) | 0.45 |
| Impaired sensory | 6% (5/84) | 7% (2/28) | 5% (3/56) | 0.74 |
| Ataxia | 12% (10/84) | 18% (5/28) | 9% (5/56) | 0.23 |
| Visual difficulties | 19% (16/84) | 14% (4/28) | 21% (12/56) | 0.43 |
| Other cranial nerve abnormalities | 6% (5/84) | 4% (1/28) | 7% (4/56) | 0.51 |
| Developmental delay | 7% (6/84) | 7% (2/28) | 7% (4/56) | 1.00 |
| Other | 32% (27/84) | 39% (11/28) | 29% (16/56) | 0.32 |
| **Positive neurological findings** | **60% (40/67)** | **64% (14/22)** | **58% (26/45)** | **0.65** |
| **Cardio-pulmonary symptoms** | **1% (1/82)** | **4% (1/27)** | **0% (0/55)** | **0.15** |
| **Gastrointestinal symptoms** | **12% (10/84)** | **14% (4/28)** | **11% (6/56)** | **0.63** |
| **General symptoms** | **28(23/83)** | **30% (8/27)** | **27% (15/56)** | **0.79** |
| **Number of symptoms, median** | **3.0 (2.0;5.0)** | **2.5 (2.0;5.0)** | **3.0 (2.0;5.0)** | **0.33** |
| **Affected laboratory values** | **11% (9/84)** | **11% (3/28)** | **11% (6/56)** | **1.0** |

Cardio-pulmonary symptoms included dyspnea, chest pain, palpitations, and others. Gastrointestinal symptoms included abdominal pain, diarrhea, vomiting, nausea, and others. General symptoms included fatigue, weight loss, and anorexia.

**Table 3. The prevalence of the neurological findings in the total cohort, the children with brain tumors and a prior musculoskeletal diagnosis (cases) versus the children with brain tumors without a prior musculoskeletal diagnosis (controls).**

| Neurological findings | Total N = 84 | Cases N = 28 | Controls N = 56 | P |
|---|---|---|---|---|
| Impaired motor or sensory function | 17% (14/84) | 14% (4/28) | 18% (10/56) | 0.68 |
| Cerebellar impairment | 4% (3/84) | 4% (1/28) | 4% (2/56) | 1.00 |
| Papillary edema | 7% (6/84) | 7% (2/28) | 7% (4/56) | 1.00 |
| Ataxia | 12% (10/84) | 28% (5/28) | 9% (5/56) | 0.23 |
| Affected vision | 8% (7/84) | 11% (3/84) | 7% (4/56) | 0.58 |
| Cranial nerve impairment | 5% (4/84) | 4% (1/28) | 5% (3/56) | 0.72 |
| Insecure gait | 5% (4/84) | 7% (2/28) | 4% (2/56) | 0.47 |
| Nystagmus | 2% (2/84) | 4% (1/28) | 2% (1/56) | 0.61 |
| Tremor | 5% (4/84) | 0% (0/28) | 7% (4/56) | 0.15 |
| Other | 13% (10/84) | 21% (6/28) | 7% (4/56) | 0.06 |

alert the physician and result in a thorough investigation, including imaging, to rule out a brain tumor [33,34].

The pre-diagnostic symptomatic interval (time from the first symptom until diagnosis) has been investigated in multiple studies over time, revealing median intervals ranging from 21 to 100 days [9,16,18,27,30,32,35–40]. In the present study, the median pre-diagnostic symptomatic interval was 87 days for the total cohort and 168 days for the subgroup with a musculoskeletal misdiagnosis, though with no statistical significance.

One might hypothesize that malignant and fast-growing tumors would increase the risk of musculoskeletal symptoms as a consequence of a higher incidence of intracranial pressure symptoms. Malignant tumors have been found in the literature with shorter pre-diagnostic symptomatic interval [26,41]. In the present study, the proportion of malignant tumors and high-grade astrocytomas was equal for the cases and controls, most likely not influencing our results. Further, the literature has revealed longer diagnostic intervals for brain tumors with a

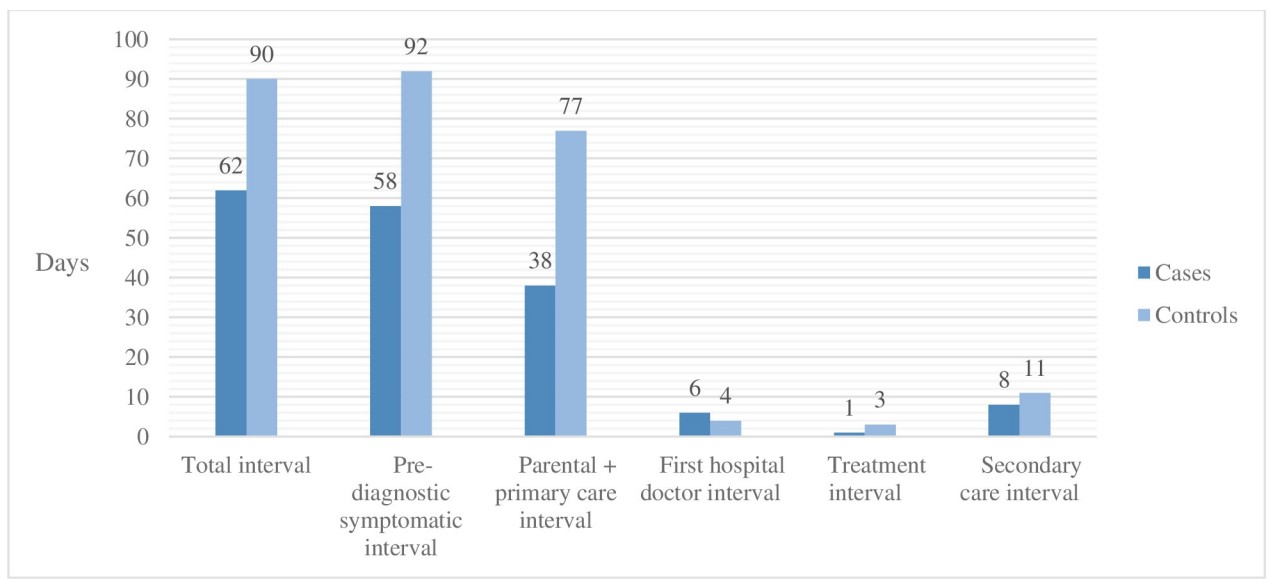

**Fig 4. Illustration of time intervals.** Bar chart illustrating six time intervals comparing the children with a brain tumor and a prior musculoskeletal diagnosis (cases) versus the children with a brain tumor without a prior musculoskeletal diagnosis (controls).

**Table 4. The prevalence of musculoskeletal diagnoses and the different time intervals comparing the children with a brain tumor and a musculoskeletal misdiagnosis to the rest of the group without a musculoskeletal misdiagnosis.**

| Diagnosis | MS misdiagnosis N = 11 | Non-misdiagnosis N = 73 | P |
|---|---|---|---|
| **Total MS diagnosis** | **100% (11/11)** | **23% (17/73)** | **<0.001** |
| Accidental MS lesion | 64% (7/11) | 6% (4/73) | <0.001 |
| Arthropathy | 0% (0/11) | 3% (2/73) | 0.58 |
| Musculoskeletal pain | 27% (3/11) | 4% (3/73) | 0.01 |
| Other bone disorder | 0% (0/11) | 3% (2/73) | 0.58 |
| Other muscle/ tendon disorder | 9% (1/11) | 3% (2/73) | 0.29 |
| Reactive arthritis | 0% (0/11) | 1% (1/73) | 0.70 |
| Scoliosis | 0% (0/11) | 6% (4/73) | 0.43 |
| **Total interval, days (IQR)** | **169 (57–582), n = 11** | **86 (32–397) n = 65** | **0.38** |
| Parental and primary care interval | 111 (21–283) n = 11 | 43 (15–222) n = 64 | 0.39 |
| First hospital doctor interval | 35 (6–166) n = 11 | 3 (1–48) n = 65 | 0.07 |
| Specialist interval | 0 (0–0) n = 11 | 0 (0–0) n = 65 | 0.40 |
| Treatment interval | 1 (1–9) n = 11 | 3 (0–7) n = 66 | 0.91 |
| Secondary care interval | 36 (7–217) n = 11 | 8 (2–76) n = 65 | 0.18 |
| Pre-symptomatic diagnostic interval | 168 (56–531) n = 11 | 75 (25–396) n = 63 | 0.27 |

MS: Musculoskeletal.

IQR: Interquartile range.

Six children with neurofibromatosis and one child with tuberous sclerosis were excluded from the analysis of time intervals. Due to lack of information in the chart review parental interval, primary care interval, and total diagnostic interval were left out in the table.

17/28 cases with a prior musculoskeletal diagnosis were not misdiagnosed. They no longer had musculoskeletal symptoms at time of brain cancer diagnosis and and their previous musculoskeletal diagnoses were presumed non-related to the cancer as it either disappeared spontaneously or responded to non-cancer treatment. Therefor it was defined as non-misdiagnoses.

supratentorial location [16,32,38,39]. In the present study, the tumor location did not differ between the cases and controls or for the subgroup with musculoskeletal misdiagnosis compared to those without, making it less likely to impact the results.

Despite medical advances, the diagnostic time intervals from symptom debut to diagnosis are still very long [16,18,27,28,30,32,35,37,38] and many brain tumors are undetected and misdiagnosed [18,19,28]. Many associations to extended diagnostic time intervals have been found (age, symptoms, and tumor grade, among others) [16,32,38] though most literature found no association between longer time intervals and decreased survival [42,43]. A musculoskeletal diagnosis prior to brain tumor diagnosis did not show significant association to influence survival in the present study, which is in accordance with the survival analysis performed in the registry-based study by Brix et al., including the total cohort of 931 children with brain tumors [20].

The number of contacts to medical care prior to the diagnosis of brain tumors has also been investigated, revealing a higher attendance in primary care one entire year preceding their cancer diagnosis when compared to children without cancer [44]. The reported number of attendances in primary and secondary care found in the literature was 4.6 (range 1–12) and 3 (0–12), respectively [29,32]. We have previously shown a median of 1 (range 1–7) hospital contacts with musculoskeletal diagnoses prior to the brain tumor diagnosis for the children with brain tumors and a prior musculoskeletal diagnosis [20].

According to the registry-based data of this study, musculoskeletal diagnosis occurred in 4% (33/931) of children with brain tumors [20], adjusted to 3% (28/931) after reviewing

medical records. Civino et al. evaluated 1277 children with cancer, in which 5% (5/101) of the children with CNS tumors had musculoskeletal symptoms, including four with arthropathy and one child with other musculoskeletal symptoms [21]. In the present study, we did not find arthropathy as a presenting manifestation in brain tumors.

This study has several limitations and must be interpreted with some caution. Recall bias may be a concern, especially regarding the time intervals and symptoms. One potential methodological limitation is the fact that the data does not include primary care and presumably underestimates the number of preliminary musculoskeletal diagnoses. Musculoskeletal pain is a common and non-specific symptom, affecting 10 to 20% of school-age children, and in primary care, musculoskeletal symptoms constitute 15% of all the contacts [45,46]. Furthermore, the estimated prevalence of registered musculoskeletal misdiagnoses of 1.2% (11/931) is presumably underestimated as a misdiagnosis. Though we tried to uncover this to the best of our ability by matching it with two gender- and age-matched controls without a registered musculoskeletal diagnosis per case. No musculoskeletal misdiagnoses occurred in the control group. Socioeconomic status was not investigated in this study and could possibly have impacted the time intervals, but since Denmark has free and equal access to medical care, this issue is considered of minor importance [47]. Another Danish study did not find socioeconomic status to influence the diagnostic delay looking at all childhood cancers [5].

Significant strengths of this study include the fact that the cases were identified from a non-selected cohort containing all consecutive cases of childhood brain tumors with a prior musculoskeletal diagnosis over 23 years with no loss to follow-up. The patients were identified through the national registries DCCR and DNPR, accompanied by detailed data from case records regarding the clinical presentation and time intervals. Furthermore, the design with randomly selected age- and gender-matched controls secured a comparison group to evaluate the clinical presentation, time intervals, and survival. Finally, the evaluation of diagnostic delay was strengthened by using a standardized model with several subintervals, which increases the generalizability of the results.

In conclusion, the present findings suggest that musculoskeletal misdiagnoses in childhood brain tumors are rare, and this study did not find a significant association to musculoskeletal misdiagnosis affecting the diagnostic delay. We highlight torticollis-related symptoms in childhood brain cancers, due to a significant risk of misdiagnosis.

## Supporting information

**S1 Checklist. STROBE statement—Checklist of items that should be included in reports of** *case-control studies.*
(DOC)

**S1 File.**
(DTA)

## Acknowledgments

We wish to thank Mette Nørgaard for the epidemiological help regarding the register data. We acknowledge Niels Henrik Bruun for statistical assistance. The authors further acknowledge Michael Thude Callesen for cooperation with data collection in the south region of Denmark.

## Author Contributions

**Conceptualization:** Laura Hallundbæk, Ninna Brix.

**Data curation:** Laura Hallundbæk, Ninna Brix.

**Formal analysis:** Laura Hallundbæk, Ninna Brix.

**Funding acquisition:** Ninna Brix.

**Investigation:** Laura Hallundbæk, Ninna Brix.

**Methodology:** Laura Hallundbæk, Jesper Amstrup, Ninna Brix.

**Project administration:** Ninna Brix.

**Supervision:** Søren Hagstrøm, Rene Mathiasen, Troels Herlin, Henrik Hasle, Kathrine Synne Weile, Ninna Brix.

**Writing – original draft:** Laura Hallundbæk, Ninna Brix.

**Writing – review & editing:** Søren Hagstrøm, Rene Mathiasen, Troels Herlin, Henrik Hasle, Kathrine Synne Weile, Jesper Amstrup, Ninna Brix.

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
