## [Decision Letter · Decision Letter 0]

11 Jan 2023

PONE-D-22-33822Musculoskeletal misdiagnoses in children with brain tumors A nationwide, multicenter case-control studyPLOS ONE

Dear Dr. Brix,

Thank you for submitting your manuscript to PLOS ONE. After careful consideration, we feel that it has merit but does not fully meet PLOS ONE’s publication criteria as it currently stands. Therefore, we invite you to submit a revised version of the manuscript that addresses the points raised during the review process. Please submit your revised manuscript by Feb 25 2023 11:59PM. If you will need more time than this to complete your revisions, please reply to this message or contact the journal office at plosone@plos.org. Please include the following items when submitting your revised manuscript:A rebuttal letter that responds to each point raised by the academic editor and reviewer(s). You should upload this letter as a separate file labeled 'Response to Reviewers'.A marked-up copy of your manuscript that highlights changes made to the original version. You should upload this as a separate file labeled 'Revised Manuscript with Track Changes'.An unmarked version of your revised paper without tracked changes. You should upload this as a separate file labeled 'Manuscript'.

We look forward to receiving your revised manuscript.

Kind regards,

Sethu Thakachy Subha, M.S

Academic Editor

PLOS ONE

2. Please ensure that you have specified (1) whether consent was informed and (2) what type you obtained (for instance, written or verbal, and if verbal, how it was documented and witnessed). If your study included minors, state whether you obtained consent from parents or guardians. If the need for consent was waived by the ethics committee, please include this information.

Reviewers' comments:

Reviewer's Responses to Questions

**Comments to the Author**

1. Is the manuscript technically sound, and do the data support the conclusions?

Reviewer #1: Partly

Reviewer #2: Yes

2. Has the statistical analysis been performed appropriately and rigorously? 

Reviewer #1: No

Reviewer #2: Yes

3. Have the authors made all data underlying the findings in their manuscript fully available?

Reviewer #1: Yes

Reviewer #2: Yes

4. Is the manuscript presented in an intelligible fashion and written in standard English?

Reviewer #1: Yes

Reviewer #2: Yes

5. Review Comments to the Author

Reviewer #1: General comments:

This study examined whether prior musculoskeletal diagnoses in childhood brain cancer are misdiagnosed and affect diagnostic delay. The topic and nationwide medical chart data are interesting. However, the major limitations of this study include a small sample size, which cannot draw a firm conclusion for observed non-statistical differences. Below are comments about this manuscript.

Major comments:

#1. How did the authors manipulate the year of diagnoses in the matching process? Because the study period extended 21 years from 1996 to 2018, the diagnostic procedures/modalities and treatments must have advanced. If comparing cases in the earlier cohort to later controls without considering diagnostic periods, confounding of those medical advances may arise.

#2. The discussion and conclusion sections, including the abstract and main text, should be cautious about concluding a strong interpretation, such as “musculoskeletal misdiagnoses in childhood brain tumors do not affect the diagnostic delay.” One primary reason for this non-significant result is the weak statistical power. In fact, in Fig 3 and Fig 4, the results suggest that cases tended to have better survival and total interval outcomes. For safer to say that this study did not find a significant association. With insufficient statistical power, please be advised to tone down such decisive conclusions/messages.

Minor comments:

#1.Page 6, Result section, the first paragraph: “From January 1996 … 46% (39/84) were girls.” This whole paragraph may be moved to the Method section (probably, in between the Data collection and Statistical analysis parts), with some subheadings such as “Cases and controls.” Because in the Method section, details for the matching process should be described, which helps readers understand the design easily.

Reviewer #2: Thank you to the editor for the opportunity to review this interesting manuscript entitled “Musculoskeletal misdiagnoses in children with brain tumors:

A nationwide, multicenter case-control study”

It is important to get so called negative findings to published, too.

Regarding abstract: I would suggest to add to results the finding of “7-fold risk of musculoskeletal symptoms in infratentorial tumors”. This is given on page 13.

In introduction, second row:… overall survival rate has shown…”. Please add survival rate of brain tumors in Denmark. This because you only refer to the Danish data (ref 3).

On row 5 about diagnostic delay you may consider to add a recent reference from Swedish register data (Rask o et al, Pediatr Blood Cancer. 2022 Nov;69(11):e29850)

About study designs, could you please provide reasoning for choosing only 2 control patients per each case as you, however, had more than 900 patients with BT-diagnosis.

In discussion, you could add the data on Swedish patients regarding the time intervals (even though no big discrepancies seem to be evident).

The strengths and limitations of this study are well discussed.

Tables and figures seem to be appropriate and their headings, too.

6. PLOS authors have the option to publish the peer review history of their article (what does this mean?). If published, this will include your full peer review and any attached files.

Reviewer #1: No

Reviewer #2: No

---

## [Author Response · Author response to Decision Letter 0]

4 Mar 2023

We hereby submit our revised manuscript PONE-D-22-33822 “Musculoskeletal misdiagnoses in children with brain tumors A nationwide, multicenter case-control study”. 

We would like to thank you and the reviewers for your time, the careful examination, and comments of the manuscript. Below all the comments addressed has been responded separately. Point-by-point responses are written in italic. Changes in the manuscript are written in red text-typ in the “Main document – marked Copy”. Please find the details below. 

Response to comments from reviewers

Reviewer 1: 

Reviewer #1: General comments:

This study examined whether prior musculoskeletal diagnoses in childhood brain cancer are misdiagnosed and affect diagnostic delay. The topic and nationwide medical chart data are interesting. However, the major limitations of this study include a small sample size, which cannot draw a firm conclusion for observed non-statistical differences. Below are comments about this manuscript.

Major comments:

#1. How did the authors manipulate the year of diagnoses in the matching process? Because the study period extended 21 years from 1996 to 2018, the diagnostic procedures/modalities and treatments must have advanced. If comparing cases in the earlier cohort to later controls without considering diagnostic periods, confounding of those medical advances may arise.

Response: Thank you for your considerations, which are appreciated. You are right about the treatment and procedure changing over time. The cases and controls were equally distributed over time. We added the following to the results, page 7: The prevalence of the cases and controls did not differ when dividing into the two periods, as 64% of the cases and 61% of the controls were diagnosed in the period 1996-2006 (p = 0.75) and 39% cases and 36% controls were diagnosed from 2007-2018. 

#2. The discussion and conclusion sections, including the abstract and main text, should be cautious about concluding a strong interpretation, such as “musculoskeletal misdiagnoses in childhood brain tumors do not affect the diagnostic delay.” One primary reason for this non-significant result is the weak statistical power. In fact, in Fig 3 and Fig 4, the results suggest that cases tended to have better survival and total interval outcomes. For safer to say that this study did not find a significant association. With insufficient statistical power, please be advised to tone down such decisive conclusions/messages.

Response: The conclusions have been toned down in the abstract, discussion and conclusion

Minor comments:

#1.Page 6, Result section, the first paragraph: “From January 1996 … 46% (39/84) were girls.” This whole paragraph may be moved to the Method section (probably, in between the Data collection and Statistical analysis parts), with some subheadings such as “Cases and controls.” Because in the Method section, details for the matching process should be described, which helps readers understand the design easily.

Response: The section has been moved, and a new introduction to the result section has been added

Reviewer #2: Thank you to the editor for the opportunity to review this interesting manuscript entitled “Musculoskeletal misdiagnoses in children with brain tumors: A nationwide, multicenter case-control study”

It is important to get so called negative findings to published, too.

Regarding abstract: I would suggest to add to results the finding of “7-fold risk of musculoskeletal symptoms in infratentorial tumors”. This is given on page 13.

Response: A sentence of this result has been added to the abstract

In introduction, second row:… overall survival rate has shown…”. Please add survival rate of brain tumors in Denmark. This because you only refer to the Danish data (ref 3).

Response: We have added this. 

On row 5 about diagnostic delay you may consider to add a recent reference from Swedish register data (Rask o et al, Pediatr Blood Cancer. 2022 Nov;69(11):e29850)

Response: the reference has been added

About study designs, could you please provide reasoning for choosing only 2 control patients per each case as you, however, had more than 900 patients with BT-diagnosis.

Response: We choose to include two controls per case as we hereby were able to secure controls being both age- and gender-matched and diagnosed with CNS tumors in the same period of time (+/- one year). Further, we do not believe that a higher number of controls would have changed the essence of the results. 

In discussion, you could add the data on Swedish patients regarding the time intervals (even though no big discrepancies seem to be evident).

Response: The reference has been added. 

The strengths and limitations of this study are well discussed.

Tables and figures seem to be appropriate and their headings, too.

Response: The headings has been altered. The figure titles has been made bold. The corresponding author line has been adjusted. Acknowledgement has been moved to after discussion and conclusion

2. Please ensure that you have specified (1) whether consent was informed and (2) what type you obtained (for instance, written or verbal, and if verbal, how it was documented and witnessed). If your study included minors, state whether you obtained consent from parents or guardians. If the need for consent was waived by the ethics committee, please include this information.

Response: a sentence declaring that formal consent is not required for the study has been added in the ethics section. 

Response: the financial disclosure section has been altered to match the section of funding information. 

Response: the sentence has been removed since the data is accessible.

The manuscript was sent back once more with the following note: 1. We note that the grant information you provided in the ‘Funding Information’ and ‘Financial Disclosure’ sections do not match.

Response: The funding information and financial disclosure text is as follows, which has also been added to the cover letter as requested: The study was funded by the Arvid Nilsson’s foundation(1780631), Børnecancerfonden (2017-1945 and 2020-6653), Ølufgard Memorial Fund (25734), and Aarhus University. No sponsors or funders (other than the named authors) played any role in study design, data collection and analysis, decision to publish or preparation of the manuscript. The authors declare no conflicts of interest.

---

## [Decision Letter · Decision Letter 1]

2 Jun 2023

Musculoskeletal misdiagnoses in children with brain tumors: A nationwide, multicenter case-control study

PONE-D-22-33822R1

Dear Dr. Brix,

We’re pleased to inform you that your manuscript has been judged scientifically suitable for publication and will be formally accepted for publication once it meets all outstanding technical requirements.

Kind regards,

Sethu Thakachy Subha, M.S

Academic Editor

PLOS ONE

Additional Editor Comments (optional):

Reviewers' comments:

Reviewer's Responses to Questions

**Comments to the Author**

1. If the authors have adequately addressed your comments raised in a previous round of review and you feel that this manuscript is now acceptable for publication, you may indicate that here to bypass the “Comments to the Author” section, enter your conflict of interest statement in the “Confidential to Editor” section, and submit your "Accept" recommendation.

Reviewer #1: (No Response)

Reviewer #2: All comments have been addressed

Reviewer #3: All comments have been addressed

2. Is the manuscript technically sound, and do the data support the conclusions?

Reviewer #1: (No Response)

Reviewer #2: Yes

Reviewer #3: Partly

3. Has the statistical analysis been performed appropriately and rigorously? 

Reviewer #1: (No Response)

Reviewer #2: Yes

Reviewer #3: I Don't Know

4. Have the authors made all data underlying the findings in their manuscript fully available?

Reviewer #1: (No Response)

Reviewer #2: Yes

Reviewer #3: Yes

5. Is the manuscript presented in an intelligible fashion and written in standard English?

Reviewer #1: (No Response)

Reviewer #2: Yes

Reviewer #3: No

6. Review Comments to the Author

Reviewer #1: I thank the authors. All concerns raised in the previous version have been revised appropriately—no further comments.

Reviewer #2: Thank you for the responses.

two minor comments:

page 5 under title cases and controls, the first sentence does not read fluently. Pleae check the verbs/ word orders.

page 7, table 2, check that all %-marks are in place

Reviewer #3: After a thorough review of the manuscript I strongly suggest some additional major comments should get addressed and a number of central points would benefit from further clarification.

Introduction

The authors state, that they aimed to “identify any patterns or red flags” in terms of the musculoskeletal misdiagnosis. However, throughout the entire manuscript it was never touched on “the red flags” again – a brief discussion on this part of the aims was entirely lacking.

Methods

The musculoskeletal diagnoses were classified according to ICD-10. It would be however very helpful to also clarify how the brain tumours have been classified (ICCC-3, ICD-10?).

Results

Indeed, the differences between misdiagnosed vs non-misdiagnosed cases did not reach statistical significance (table 4), I suggest, however, to mention the tendency of larger time intervals among misdiagnosed children in the results and discussion section.

The authors state only one odds ratio in the results section. I suggest adding a table or a figure where the all odds ratios are transparently displayed. This would be very helpful for a better understanding of the associations between potential risk factors and the musculoskeletal misdiagnosis.

Discussion

Although the low sample size hardly allows for statistically significant findings, the authors may consider carefully those values indicating potential deviations in cases and controls. Therefore, I wondered whether the authors may add a paragraph where they discuss their own findings and speculate on potential underlying mechanisms possibly driving their findings, e.g. longer time intervals among misdiagnosed children or why the proportion of sequelae was lower among cases than controls (against expectation)

Since the small sample size of the study has clearly influenced the precision of the estimates, I suggest adding this point to the paragraph on strengths and limitations.

Figure 4:

I wondered why the number of days in the pre-diagnostic symptomatic interval in controls is higher than in the total interval (92 vs 90). It should be vice versa – shouldn’t it?

In addition: what is the meaning of the numbers – Mean, Median? I suggest to add this information to the figure caption.

General

I would ask the authors to use consistent word usage for sex/ gender

7. PLOS authors have the option to publish the peer review history of their article (what does this mean?). If published, this will include your full peer review and any attached files.

Reviewer #1: No

Reviewer #2: No

Reviewer #3: No

---

## [Editor Report · Acceptance letter]

15 Jun 2023

PONE-D-22-33822R1 

Musculoskeletal misdiagnoses in children with brain tumors:
A nationwide, multicenter case-control study 

Dear Dr. Brix:

I'm pleased to inform you that your manuscript has been deemed suitable for publication in PLOS ONE. Congratulations! Your manuscript is now with our production department. 

Kind regards, 

on behalf of

Dr. Sethu Thakachy Subha 

Academic Editor

PLOS ONE